# Complexity Matters: Effective Dimensionality as a Measure for Adversarial Robustness

## Abstract

Quantifying robustness in a single measure for the purposes of model selection, development of adversarial training methods, and anticipating trends has so far been elusive. The simplest metric to consider is the number of trainable parameters in a model but this has previously been shown to be insufficient at explaining robustness properties. A variety of other metrics, such as ones based on boundary thickness and gradient flatness have been proposed but have been shown to be inadequate proxies for robustness.

In this work, we investigate the relationship between a model's *effective dimensionality*, which can be thought of as model complexity, and its robustness properties. We run experiments on commercial-scale models that are often used in real-world environments such as YOLO and ResNet. We reveal a near-linear inverse relationship between effective dimensionality and adversarial robustness, that is models with a lower dimensionality exhibit better robustness. We investigate the effect of a variety of adversarial training methods on effective dimensionality and find the same inverse linear relationship present, suggesting that effective dimensionality can serve as a useful criterion for model selection and robustness evaluation, providing a more nuanced and effective metric than parameter count or previously-tested measures.

## 1 Introduction

Robustness to adversarial perturbations has been a desired but often ill-achieved property of modern neural networks, that can otherwise be increasingly found in commercial real-world applications (Tocchetti et al., 2022). There has been a myriad of techniques proposed to improve robustness, ranging from preproccessor defenses (Diallo & Patras, 2024) (that have since been shown to be flawed by Carlini (2024)) to various forms of adversarial training methods. Adversarial training, since its introduction by Madry et al. (2019), has become one of the most common methods of defending against adversarial examples owing to its ease of implementation and deployment. It was however shown recently that adversarial training is far from a catch-all approach and suffers from overfitting (Rice et al., 2020).

A variety of different measures have been proposed to quantify the inherent robustness of trained models, but they were mainly found to be limited in their applicability Kim et al. (2023). We investigate the use of *effective dimensionality* as a measure for adversarial robustness. This measure was originally proposed by MacKay (1992) and then expanded to deep neural networks by Maddox et al. (2020). We use effective dimensionality as a metric of model complexity – it is a joint measure of both the expressivity of the model's architecture and the inherent properties of the dataset the model was trained on. This was used as it was found that parameter count alone does not provide a sufficient metric to represent the joint 'complexity' of a model and it's underlying trained data.

We conduct experiments across a wide range of model architectures, such as ResNet (He et al., 2015), ShuffleNet (Zhang et al., 2017), and YOLO (Varghese & M., 2024), as well as over CIFAR-10, CIFAR-100, and ImageNet. Our key research contributions are as follows:

1. We present a large-scale investigation into the effective dimensionality of classification models, most of which are often used in production.

2. We find a *near-linear negatively-correlated relationship* between a model's robustness to adversarial examples and its effective dimensionality. Namely, we find that models with a *lower* effective dimensionality exhibit *improved* robustness properties.

3. We conduct an empirical investigation into the effect of adversarial training techniques on a model's effective dimensionality. We show how these techniques reduce the models' effective dimensionality and derive a linear trend in-line with the above findings.

## 2 RELATED WORK

### 2.1 ADVERSARIAL ROBUSTNESS

**Threat Model** For a given model $f_\theta : \mathcal{X} \to \mathbb{R}^C$ parameterised by $\theta$, an input $\mathbf{x} \in \mathcal{X}$ and label $y =_{\mathrm{argmax}} f_\theta(x) \in \{0, \cdots, C\}$, we consider a perturbation $\delta$ to be adversarial if

$$f_\theta(\mathbf{x} + \delta) \neq_{\mathrm{argmax}} y \tag{1}$$

Working from established literature such as Croce et al. (2021), we assume a white-box threat model for the attacker (i.e. full knowledge of both $f$ and $\theta$), and measure adversarial robustness via robust accuracy under attack for the $l_\infty$ norm. Details regarding the exact metrics used are discussed in Section 3.

**Adversarial Example Generation** Over the years, various strategies have been developed to assess and challenge the robustness of models. Among the most notable are Projected Gradient Descent (PGD) (Madry et al., 2019), AutoAttack (Croce & Hein, 2020b), and Gaussian noise (Ford et al., 2019).

PGD, introduced by Madry et al. (2019), is one of the most widely studied adversarial example generation methods due to its effectiveness and simplicity. PGD is an iterative method that builds on the idea of applying small, constrained gradient-based perturbations to an input in order to maximize the model's loss function. It is an iterative method, with the perturbation being projected back into a predefined bound around the original input to ensure the perturbation stays within acceptable limits. PGD is considered one of the strongest first-order methods and serves as the basis for adversarial training, as discussed later.

AutoAttack, proposed by Croce & Hein (2020b), introduces a suite of strong adversarial attacks. AutoAttack automates the process of adversarial example generation generation and evaluation by combining four different attacks, including PGD and Fast Adaptive Boundary (Croce & Hein, 2020a). This ensemble approach ensures comprehensive robustness evaluation by removing potential biases inherent in single-attack methods. It is designed to be parameter-free and eliminates the need for tuning, providing more consistent and robust adversarial performance testing across a variety of neural network architectures.

Ford et al. (2019) explored the use of simple Gaussian noise as a baseline adversarial perturbation. While Gaussian noise is not a crafted example in the same sense as PGD or AutoAttack, it serves as a valuable point of comparison for more sophisticated methods. Though less effective than gradient-based attacks in fooling models, Gaussian noise highlights a model's inherent robustness to random variations and provides insights into the stability of learned representations. Ford et al. (2019) demonstrated that while Gaussian noise is often ineffective in generating adversarial examples, it remains a useful tool for probing the overall robustness of models, particularly in scenarios where adversarial examples are not deliberately targeted.

**Adversarial Training** We explore the effects of *adversarial training* and other derivative methods. This robust training technique was first formulated by Madry et al. (2019), utilizing the following min-max optimization with notation similar to the one used above:

$$\min_f \mathbb{E}_{(\mathbf{x}, y) \sim D}[\max_\delta l(f(\mathbf{x} + \delta), y)] \tag{2}$$

The inner maximization is often approximated by repeated iterations of PGD, which greatly increases computation costs. Adversarial training is considered to be the state-of-the-art method for ensuring robustness, with recent papers greatly reducing involved costs by re-using gradient information during training (Shafahi et al., 2019).

## 2.2 ROBUSTNESS MEASURES

Several measures have been proposed to quantify model robustness. Boundary thickness, introduced by Yang et al. (2021), focuses on exploring the relationship between the distance between decision boundaries and data points. It was found that models with thicker boundaries tend to exhibit greater robustness, as adversarial perturbations are less likely to cross decision boundaries with small perturbations. Flatness-based measures, studied by Stutz et al. (2021), focus on the curvature of the loss surface around the input data. Models with flatter loss landscapes are thought to generalize better and resist adversarial perturbations. Yang et al. (2020) proposed the use of local Lipschitzness to assess robustness, relating to the smoothness of the learned decision function. It was found that a lower local Lipschitz constant implies greater robustness, as small perturbations induce minimal output changes. However, calculating the local Lipschitz constants directly is often computationally-intractable, and estimators need to be used which are in themselves computationally expensive on large models (Fazlyab et al., 2023).

Despite these advances, Kim et al. (2023) demonstrated that none of these metrics provide a comprehensive measure of robustness. Each captures only specific aspects, leaving room for the development of more holistic and effective metrics.

## 2.3 EFFECTIVE DIMENSIONALITY AS A COMPLEXITY METRIC

We work under the intuition that the robustness properties of a model are related to it's complexity. The issue arrises with trying to concretely define 'complexity' when it comes to neural networks. A simple measure would be model size in terms of number of trainable parameters or amount of compute required, but this has recently been shown to only have a small correlation to robustness (Debenedetti et al., 2023). Grant & Wu (2022) investigated the relationship between generalization and generalized degrees-of-freedom. We investigated this metric, but found that the method described in the paper to be generally computationally intractable and not converge for larger models.

Hence, we decided to explore the relationship between *effective dimensionality*, as defined by Maddox et al. (2020), and adversarial robustness. We found this metric to be appropriate due to it being a joint measure of both the expressivity of the model's architecture and the inherent properties of the dataset the model was trained on. We felt that this captured the notion of what a neural network's complexity should be in accordance to the universal approximation theorem (Hornik et al., 1989) :- the architecture of the model itself does not affect complexity until after the parameters are trained on, and the complexity would naturally depend on the kind of data the model was being fitted to.

The effective dimensionality of a symmetric matrix $A \in \mathbb{R}^{k \times k}$ is defined to be

$$N_{\text{eff}}(A, z) = \sum_{i=1}^{k} \frac{\lambda_i}{\lambda_i + z} \tag{3}$$

where $\lambda_i$ are the eigenvalues of $A$ and $z > 0$ is a regularization constant (MacKay, 1991). We compute the effective dimensionality of a neural network based on the eigenspectrum of the Hessian of the loss on the *test* data, according to the method described by Maddox et al. (2020). This metric has previously been shown to accurately track generalization properties of networks.

Intuitively, the effective dimensionality of a model explains the number of parameters that have been determined by the data, which corresponds to the number of parameters the model is using to make predictions. A model with a low effective dimensionality have a simpler function space, embodying Occam's razor and generally avoid overfitting. The metric is directly related to the number of parameter directions in which the functional form of the model is sensitive to perturbation (Maddox et al., 2020).

## 3 EXPERIMENTS

We conduct experiments over the CIFAR10, CIFAR100 (Krizhevsky, 2009), and ImageNet (Deng et al., 2009) datasets.

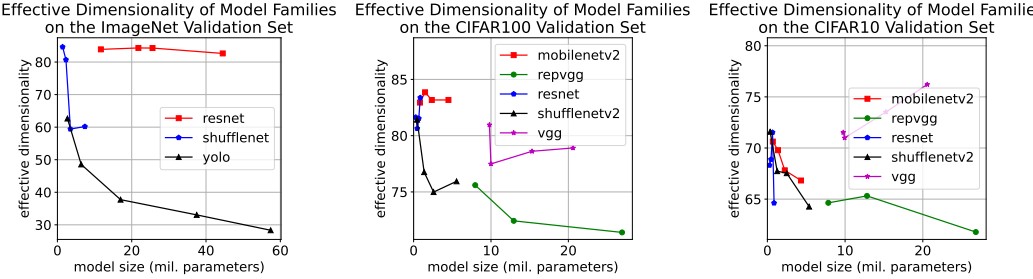

Figure 1: Measuring the effect of scale on a model's effective dimensionality, on various datasets. The exact models tested from each model family are listed in Section 3.

For ImageNet, we test the following model families: ResNet (`resnet18`, `resnet34`, `resnet50`, `resnet101`), ShuffleNetV2 (`shufflenetv2_x0_5`, `shufflenetv2_x1_0`, `shufflenetv2_x1_5`, `shufflenetv2_x2_0`), and YOLOv8 (`yolov8n-cls`, `yolov8s-cls`, `yolov8m-cls`, `yolov8l-cls`, `yolov8x-cls`).

For CIFAR10 and CIFAR100, we test the following models families: ResNet (`resnet20`, `resnet32`, `resnet44`, `resnet56`), MobileNetV2 (`mobilenetv2_x0_5`, `mobilenetv2_x1_0`, `mobilenetv2_x1_5`, `mobilenetv2_x2_0`), RepVGG (`repvgg_a0`, `repvgg_a1`, `repvgg_a2`), ShuffleNetV2 (`shufflenetv2_x0_5`, `shufflenetv2_x1_0`, `shufflenetv2_x1_5`, `shufflenetv2_x2_0`), VGG (`vgg11_bn`, `vgg13_bn`, `vgg16_bn`, `vgg19_bn`)

We conduct the following three large-scale experiments:

**Measuring effective dimensionality** We calculate the effective dimensionality of a large number of commercial-grade models on the aforementioned vision datasets. As described in Section 2.3, we use a slightly modified version of the code provided by Maddox et al. (2020) to calculate effective dimensionality.

**Robustness at different effective dimensionalities** We calculate the robustness of the various models listed above. We use the implementation provided by Kim (2020) to enable this. We run PGD, Auto, GN adversarial examples at varying increasing capacities ($\epsilon \in \{1/255, 2/255, \cdots, 8/255\}$ and $\sigma \in \{0.05, 0.1, \cdots, 0.4\}$ respectively). The robustness is reported as classification accuracy under attack $p^*$. To account for the fact that different models vary in their baseline (i.e. without any perturbations being applied) accuracy $p$, we report the *relative* performance under attack $p_r = \frac{p^*}{p}$. This gives a more practical metric for robustness that measures the relative *degradation* in performance under attack.

**Effect of adversarial training on effective dimensionality** We measure the effect of adversarial training and related techniques on effective dimensionality and relate this to the robustness properties of the respective models. We test `ResNet18, WRN28, WRN34` using the MAIR framework (Kim et al., 2023) and the techniques described there. Namely, we consider the effect of extra data (Carmon et al., 2022), and using Adversarial Weight Perturbation (AWP) (Wu et al., 2020) as an optimizer. We look at and compare various training methods: Standard, AT (Madry et al., 2019), TRADES Zhang et al. (2019), MART (Wang et al., 2020).

## 4 RESULTS

### 4.1 EFFECTIVE DIMENSIONALITY VS MODEL SIZE

In Figure 1 we demonstrated how model size, measured as the number of trainable parameters, affected the model's effective dimensionality. These results are in-line with what would be expected based on the smaller models tested by Maddox et al. (2020). With the exception of outliers `resnet` on `ImageNet`, and `vgg` on the `CIFAR` datasets, we can see a clear polynomial trend :- the effective dimensionality of the models decays as they get larger. In the outliers' cases, the effective dimen-

sionality remains more-or-less consistent or without any noticeable trend. We note in Section 4.3 that the robustness of these outliers tends to follow the trend of their effective dimensionality regardless.

## 4.2 Adversarial Performance vs Model Size

We corroborate past findings presented by Debenedetti et al. (2023); Bartoldson et al. (2024) that larger, in terms of parameter count, models tend to be more less effected by adversarial examples with a given parameter budget. This trend however does not hold in all tested model classes and is marginal at times. This marginal relationship corresponds to recent similar findings in exploring robustness in LLMs given size that found little-to-no correlation between robustness and size as in Howe et al. (2024).

We find that higher-parameterised models tend to be more robust *within the same model class*. However, it was found that there is little correlation between model size and robustness between different model classes. A more detailed discussion of these results can be found in Appendix A.1.

## 4.3 Adversarial Performance vs Effective Dimensionality

The graphs shown in Figure 2 illustrate the relationship between effective dimensionality and the relative performance of the different models (listed in Section 3) under different adversarial examples (described in Section 2.1), across multiple datasets (ImageNet, CIFAR100, CIFAR10). We observe the following trends:

**AutoAttack** Across all datasets (ImageNet, CIFAR100, CIFAR10), there is a *general negative correlation* between effective dimensionality and relative performance. Models with higher effective dimensionality tend to suffer more under AutoAttack, especially on ImageNet and CIFAR100. The main outliers to the general trends are ResNet (ImageNet) and VGG (CIFAR), much like in Section 4.1.

**PGD** The PGD results show a similar trend across ImageNet and CIFAR datasets, where models with lower effective dimensionality generally exhibit higher robustness. However, there are some inconsistencies, particularly in CIFAR100 and CIFAR10, where certain models (e.g., VGG on CIFAR datasets) exhibit an increase in performance at higher dimensionalities, likely due to model-specific behavior.

**Gaussian noise** When subject to Gaussian noise, models across all datasets show a clear inverse relationship between effective dimensionality and performance. Higher dimensionality again corresponds to lower robustness.

Overall, the results suggest that *lower effective dimensionality* generally correlates with *better adversarial robustness* across various adversarial example generation methods and datasets, though model-specific variations do exist.

## 4.4 Effect of Adversarial Training on Effective Dimensionality

In Figures 3 and 4 we demonstrate the effect of various adversarial training techniques on the effective dimensionality of models and how this relates to their adversarial robustness.

The results are incredibly clear-cut. We see that in nearly-all cases the `Standard-None` setup has the highest dimensionality, with any additional adversarial training measures significantly lowering this metric. For `ResNet18`, `AWP` reduced dimensionality by 24.0%, and `AWP+ED` by 29.3% relative to the baseline, on average. For `WRN28`, `AWP` reduced dimensionality by 23.2% and `AWP+ED` by 30.4%, on average. For `WRN34`, `AWP` reduced dimensionality by 19.1% and `AWP+ED` by 31.3%, on average.

The effect of these dimensionality reductions on adversarial robustness can be seen in Figure 4. We remove outliers (mainly poorly pre-trained models) when plotting these results. Here we see a highly-correlated (lowest $R^2$ is 0.73 in the `WRN34` case) relationship between effective dimensionality and the relative adversarial performance. Based on the linear regression, we can see that drop in effective dimesionality of 10 points corresponds to roughly an absolute increase of 5.5% in relative adversarial performance. These results are in line with the ones described in Section 4.3 above, namely that lower effective dimensionality generally correlates with higher robustness.

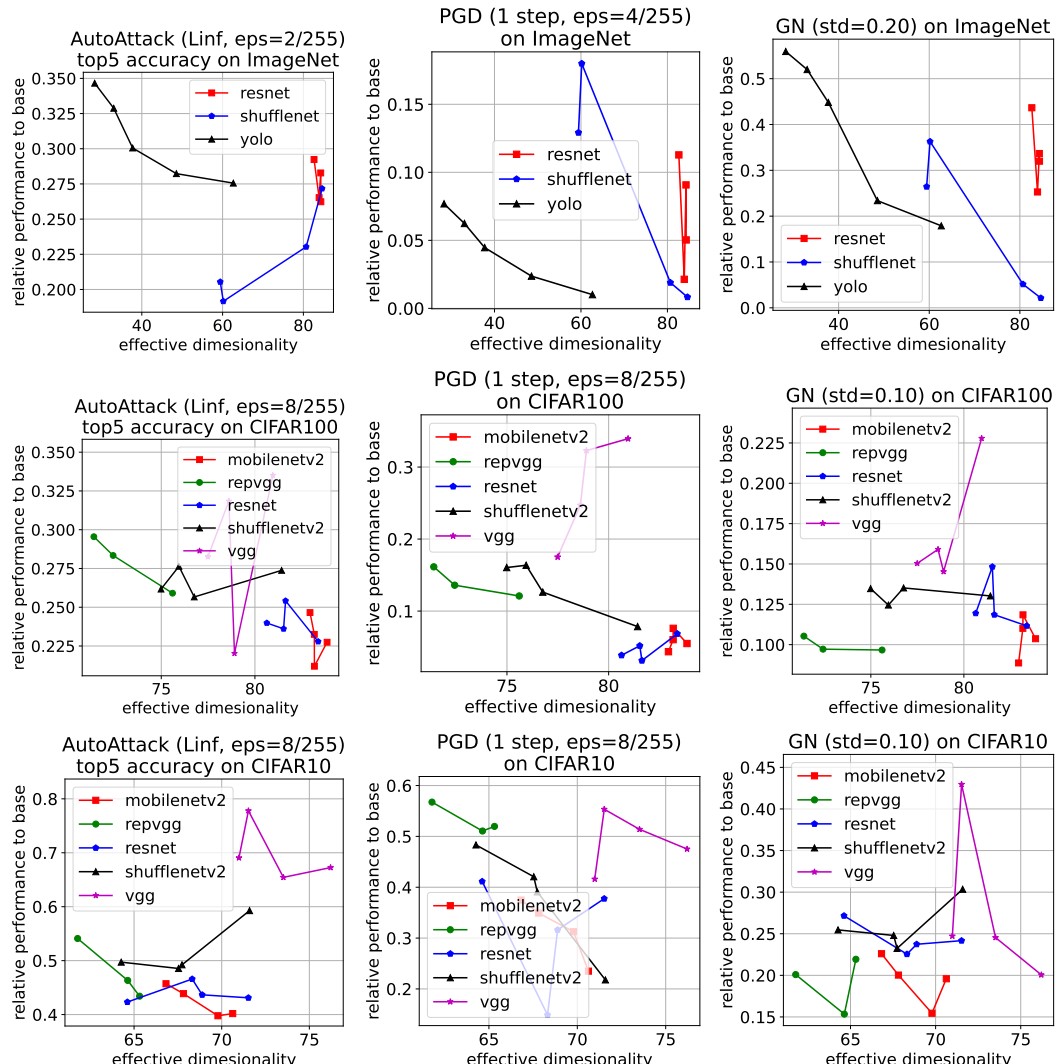

Figure 2: Relative adversarial performance, plotted against the respective model's effective dimensionality. A description of the performance metric is given in Section 3. We report the top-5 accuracy for AutoAttack and the top-1 accuracy for PGD and GN.

## 5 DISCUSSION

The results presented in Section 4 build upon the work of Maddox et al. (2020) who claim that the effective dimensionality of a model is a good mechanism for model selection. We show that this selection criteria does not only correspond to an improvement in generalization but also to adversarial robustness.

Across all experiments, a clear inverse relationship between effective dimensionality and robustness under adversarial examples emerges, consistent with previous findings that models with lower effective dimensionality tend to be more robust to adversarial perturbations. The adversarial training methods tested consistently reduced the effective dimensionality of models compared to standard training, confirming that adversarial training not only enhances robustness but also simplifies the model's internal structure in terms of its dimensionality.

Although effective dimensionality has shown promise as a proxy for measuring adversarial robustness, it is not a perfect predictor as shown by the slightly mixed results presented in Section 4.3. Other factors, such as the loss landscape flatness, boundary thickness, or specific adversarial train-

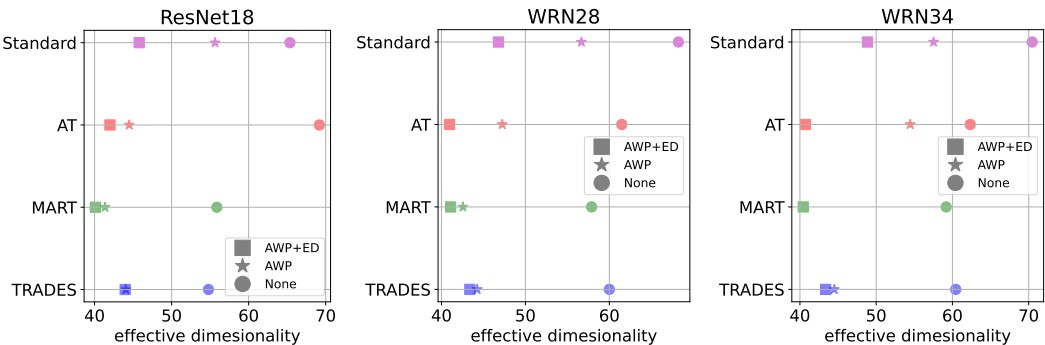

Figure 3: Effect of various adversarial training methods, described in Section 3, on the effective dimensionality of the respective models. `AWP` corresponds to Adversarial Weight Perturbation, and `AWP+ED` involves AWP and extra training data.

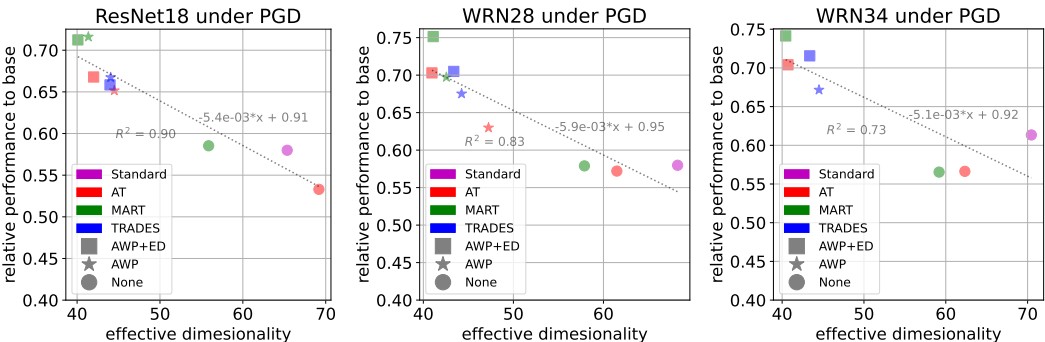

Figure 4: Relative adversarial performance under a various adversarial training methods, plotted against the respective model's effective dimensionality. A description of the performance metric is given in Section 3. `AWP` corresponds to Adversarial Weight Perturbation, and `AWP+ED` involves AWP and extra training data.

ing mechanisms, may also contribute significantly to the model's ability to withstand adversarial examples. This aligns with the conclusion of Kim et al. (2023) that no single robustness measure is comprehensive, leaving space for the development of more holistic metrics.

## 5.1 LIMITATIONS

Several important limitations must be noted. First, due to resource constraints, we were unable to test larger more complex models. As a result, our findings are limited to smaller model architectures, and it remains unclear whether the observed trends hold for larger models, which could exhibit different behaviors with respect to dimensionality and robustness.

Additionally, the results are based purely on empirical analysis, and while we observed consistent patterns, these findings do not establish a direct causal relationship between effective dimensionality and adversarial robustness. For example, there were clear outliers present in Section 4.3 and Section 4.1. This points to the need for further theoretical exploration to better understand the underlying mechanisms and interactions between dimensionality and other factors, such as the loss landscape or boundary geometry.

## 5.2 FUTURE WORK

We conducted an extensive investigation into how model scale and different adversarial training methods affect effective dimensionality, and how this relates to the models' adversarial performance. However, it will be worthwhile to conduct an evaluation into how model quantization and distillation affect this measure in relation to robustness. A future branch of work would also involve conducting a similar exploration for other architectures, such as vision transformers, and on different domains, such as reinforcement learning environments.

Given the observed trends, future research could explore the integration of multiple robustness metrics, including effective dimensionality, boundary geometry, and loss flatness, to develop a more unified and accurate predictor of robustness. It would also be worthwhile to explore why certain models (such as ResNet and VGG, mentioned in Section 4.1) are outliers from this observed trend.

## 6 CONCLUSION

In this paper, we presented an extensive empirical investigation into the relationship between effective dimensionality and adversarial robustness in deep neural networks. Through our experiments on production-scale models, including YOLO and ResNet architectures, we demonstrated that effective dimensionality serves as a strong predictor of robustness to adversarial examples. Specifically, we found a linear inverse correlation between the two, showing that models with lower effective dimensionality tend to exhibit greater robustness. This trend holds both within architecture families, where larger models generally possess lower complexity, and under adversarial training techniques, which further reduce effective dimensionality in line with improved robustness.

These findings suggest that effective dimensionality can serve as a useful criterion for model selection and robustness evaluation, providing a more nuanced and effective metric than parameter count or existing flatness- and boundary-based measures. However, our study is limited to empirical observations, and further theoretical work is required to fully understand the mechanisms driving the relationship between complexity and robustness. Nonetheless, our work lays a foundation for future research on the role of effective dimensionality in adversarial robustness and model optimization.

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

## A APPENDIX

### A.1 ADVERSARIAL PERFORMANCE VS MODEL SIZE

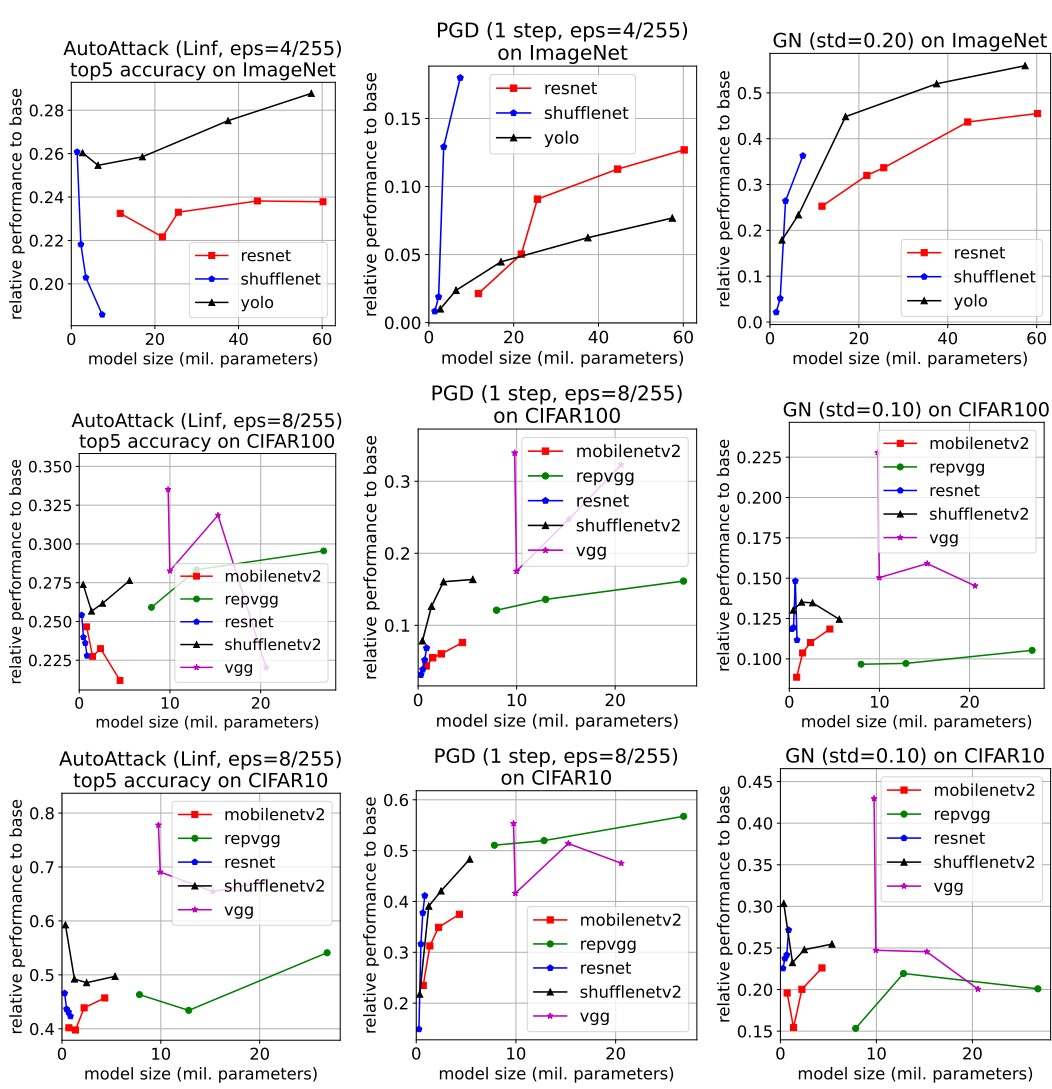

Figure 5: Relative adversarial performance, plotted against the respective model's size, measured in number of trainable parameters. A description of the performance metric is given in Section 3. We report the top-5 accuracy for AutoAttack and the top-1 accuracy for PGD and GN.

