# OpenReview forum: "Complexity Matters: Effective Dimensionality as a Measure for Adversarial Robustness"
_ICLR.cc/2025/Conference — ICLR 2025 Conference Withdrawn Submission_

### Official Review · Reviewer_Deu1 · 2024-11-01

**Soundness:** 1
**Presentation:** 2
**Contribution:** 2
**Rating:** 3
**Confidence:** 4

**Summary:**

The paper links adversarial robustness to the models' effective dimensionality, using empirical experiment to investigate the relationship between these two aspect of the model.  The paper perform various experiments on different datasets (e.g. CIFAR10, CIFAR100, ImageNet) across different model architectures (e.g. ResNet, ShuffleNet, Yolo) and tries the prove the negative relationship between effective dimensionality and adversarial robustness. However, empirical results cannot validate such claim.

**Strengths:**

1. The paper investigates the relationship between two metrics, models' effective dimensionality and adversarial robustness. Various empirical results are presented, which are helpful for readers to get the relationship of two these results.

2. The paper is generally easy to read.

**Weaknesses:**

1. The definition of effective dimensionality is highly correlated to the adversarial accuracy itself. Actually, it seems to be a tautology of adversarial robustness. effective dimensionality measures the flatness of loss landscape on the test data using Hessian Matrix while adversarial robustness reflects the flatness of loss landscape using the accuracy.

2. The empirical result cannot prove the claim of the paper. As shown in Figure 2, the relationship between effective dimensionality and adversarial robustness is chaotic without clear trend.

**Questions:**

See above.

---

### Official Review · Reviewer_xK68 · 2024-11-03

**Soundness:** 3
**Presentation:** 2
**Contribution:** 1
**Rating:** 3
**Confidence:** 4

**Summary:**

In this paper, the authors have investigated the relationship between the effective dimensionality and the model robustness, where the effective dimensionality shows a negative linear correlation with the model robustness.

**Strengths:**

**Strength 1:** This paper provides the empirical findings to understand the robustness capability of various deep model architectures.
- Evaluating and understanding the robustness of deep models, or the generalization capability with a wide viewpoint, are long-lasting issues crucial in the theoretical understanding of the parametric model's capability and in realizing robust AI models in applications. From this perspective, the main strength of this paper is that it provides a novel empirical finding to understand model robustness.

**Weaknesses:**

**Weakness 1:** Limited theoretical understanding of the effective dimensionality
- This paper only investigates the empirical results to show the relationships between the effective dimensionality and the model robustness, but its theoretical understanding is not yet explored.
- Specifically, to further provide in-depth understanding, it would be better to explain why the effective dimensionality correlates more clearly with the model robustness than other measurements.

**Questions:**

**Question 1:** Any conjecture to explain the clear correlations between the effective dimensionality and the model robustness
- Would you explain why clear relationships are observed, which is not shown in the prior metrics?

**Question 2:** Relationships between the effective dimensionality and the model flatness
- In Eq. 3, the effective dimensionality seems to be strongly related to the eigenvalues of Hessian. When revisiting the traditional measurements of model flatness on the parameter space, i.e., the maximum or summation of the eigenvalues of Hessian, the effective dimensionality increases when the model flatness decreases. Thus, it is hard to figure out why the effective dimensionality is particularly showing the more clear relationships to the model robustness. Would you share your thoughts on the relationships between the effective dimensionality and the model flatness?

---

### Official Review · Reviewer_3nZ2 · 2024-11-03

**Soundness:** 1
**Presentation:** 2
**Contribution:** 1
**Rating:** 3
**Confidence:** 4

**Summary:**

The authors evaluate adversarial robustness and effective dimension on an array of models of different architectures, trained with different datasets, and trained with various adversarial training algorithms. The authors plot the relationship between robustness and effective dimension and make four claims: (1) Effective dimension sometimes decreases with model size; (2) adversarial robustness is correlated with model size within the same model class, but not outside; (3) adversarial robustness and effective dimension are negative correlated; (4) adversarial training reduces effective dimension.

**Strengths:**

* The paper is generally well-written and clear.
* The paper addresses and important topic: identifying factors that either are causally related to, or correlate very strongly with adversarial robustness is an important area of research, for cheap model selection or for understanding how to train adversarially robust models.
* The ideas proposed by the paper are novel, to my knowledge.

**Weaknesses:**

The primary weakness of this paper is major: the evidence presented in this paper does not sufficiently support the claims made by the authors. The primary premise of the paper is built by the following claims made by the authors: (1) adversarial robustness and effective dimension are negatively correlated; (2) effective dimension decreases with adversarial training.
1. The negative correlation between adversarial robustness and effective dimension is weak at best, and often yields the opposite trend. The evidence for this claim is given in Figure 2. In this figure, ResNet appears to clearly have the claimed trend in 1 out of 9 plots (bottom right only). ShuffleNetV2 has the opposite trend in 2 out of 9 plots, and appears to have no correlation in another 2 out of 9, leaving only 5 out of 9 plots with the claimed trend. VGG seems to have the opposite trend or appears noisy in all plots in which it appears.  RepVGG has the opposite trend in the bottom right trend. MobileNetV2 seems to have the opposite trend, or noise in 4 out of the 6 plots in which it appears (all but the bottom left two plots). In total, the only model that actually observes the claimed trend is YOLO, which only appears in three of the plots.
2. The fact that adversarial training decreases effective dimensionality has slightly stronger, but still not sufficient evidence: the adversarial training in Figure 3 left has greater effective dimensionality than the standard training. Further, the relationship between effective dimensionality and robustness does not appear correlated when considered training without AWP in two of the three plots in Figure 4 (center and right, considering the circle points). It seems, therefore, that perhaps AWP leads to both substantially smaller effective dimension and also better robustness, but the claim made by the authors, that robustness in general is negative correlated with effective dimensionality, appears false.

Related work:
* AWP is closely related to Sharpness-Aware Minimization, and so it would be beneficial to include some discussion of the relationship between Sharpness-Aware Minimization and effective dimension. For example, see [1].

Minor:
* The clarity of your plots could be improved by making the legends not overlap with the plotted lines, and by keeping colors consistent across nearby plots (e.g., the models are given different colors in different plots in Figure 2)

[1] Andriushchenko, Maksym, et al. "Sharpness-aware minimization leads to low-rank features." Advances in Neural Information Processing Systems 36 (2023): 47032-47051.

**Questions:**

See weaknesses.

---

### Official Review · Reviewer_kPjZ · 2024-11-04

**Soundness:** 3
**Presentation:** 3
**Contribution:** 2
**Rating:** 3
**Confidence:** 3

**Summary:**

The authors explore the relationship between "effective dimensionality" and robustness for a collection of computer vision classification models.

**Strengths:**

The paper is relatively clearly written.

The question being asked and answered is compelling.

**Weaknesses:**

* the paper is quite short and sweet: some experiments are conducted, on small models, on a single seed.
* there is no "novel" contribution from the paper. In itself, this is not a problem at all. But without a novel contribution, I would expect the analysis to be more in-depth (not being able to run larger models is understandable; perhaps running more than one seed and more models is feasible? Or some of the other directions that the authors mention could be explored: looking at other architectures or boundary geometry or loss flatness. Or more exploration of what happens during adversarial training.
* the paper is too superlative in its writing, almost to make up for interesting but not the strongest-looking results (which is also ok, just means more investigation is needed).
* overall it seems like valuable work to be presented at a computer vision or robustness workshop, but the contribution is not enough to meet the ICLR main conference bar in my current opinion
* handful of typos

**Questions:**

## Questions
* 149: Calculating the hessian is generally super expensive. Is there some trick here to make it scale? Or does this only work on small models?
* 215: "clear polynomial trend" I think this is significantly too generous -- there appears to be a lot of bumpiness in the plots. Do you know what's going on there? Can you get more seeds?
* 247: it seems much too strong to say that "models across all datasets" show a clear inverse trend when several of the lines go up and not down in the right column of figure 2 plots. Am I misunderstanding something?
* can you get more seeds? it's hard to tell what's going on with only one seed.
* I don't really understand what's going on with the adversarial training setup and plots (figure 3). Maybe you can give more details on the specific of adversarial training? I think there could be interesting details to look through here.

## Specific things:
* line 34: "has been a myriad of" -> "have been myriad"
* line 35: isn't postprocessing an important thing too?
* 39: delete "recently"
* 48: it's -> its
* 53: what does "most of which are often used in production" mean?
* 66: please make it obvious you're in the computer vision setting
* 68: delete the argmax
* 72: use \ell instead of l
* 74-99: why these three methods? there are many many adversarial example finding techniques. why did you choose these specifically? why no FGSM as a baseline?
* 103: please define what is in the equation; people won't have necessarily read the previous papers
* 106: I think it's generous to say adversarial training is SOTA. it's a go-to technique and important tool, but it alone will not get you SOTA robustness
* 129: it's -> its
* 173: exact -> specific
* 212: demonstrated -> show
* 215: the ":-" is not something I've seen before and I don't know what it means
* 233: graphs -> plots
* 258-263: writing it out like this is hard to follow. make a table and highlight one important result from it if you want?
* 265: "we removed outliers" maybe you can find a different way to include the outliers (maybe make a scatter but don't include them in the fit and make it obvious by using a different color?). Removing data (which I'm grateful you mentioned in the text) raises suspicions of cherry-picking.
* 269: delete "above"
* 323: slightly mixed -> mixed
* 380: extensive investigation -> investigation
* 393: same
* 395: you mention production-scale models here, but previously you just said you had to do only small models because of computational constraints. I would expect either that these models are production-scale and that's ok, or that you don't have the compute for production-scale and that's ok, but not both?

## General note
With 3 seeds per model (possibly even only for one family of models and one adversarial attack method) this could be a great workshop paper. As is, I can't recommend it for main conference. I could imagine increasing my rating to a 5 if more seeds are run, all the data are shown, clearer justification is given to attack methods, and the results look less noisy OR are talked about in a less superlative way. Going deeper into the adversarial training direction could be quite interesting and would start to look more like a main conference paper to me, though I imagine that might be infeasible to do in the rebuttal time-window.

---

### Note · Authors · 2024-11-20

I have read and agree with the venue's withdrawal policy on behalf of myself and my co-authors.